# The Respiratory Management of the Extreme Preterm in the Delivery Room

**DOI:** 10.3390/children10020351

**Published:** 2023-02-10

**Authors:** Raquel Escrig-Fernández, Gonzalo Zeballos-Sarrato, María Gormaz-Moreno, Alejandro Avila-Alvarez, Juan Diego Toledo-Parreño, Máximo Vento

**Affiliations:** 1Department of Neonatology, Hospital Universitari i Politècnic La Fe, 106 Fernando Abril Martorell Avenue, 46026 Valencia, Spain; 2Department of Neonatology, Hospital Gregorio Marañón, 28009 Madrid, Spain; 3Division of Neonatology, Pediatric Department, Complexo Hospitalario Universitario de A Coruña (CHUAC), Sergas, 15006 A Coruña, Spain

**Keywords:** prematurity, fetal-to-neonatal transition, non-invasive ventilation, oxygenation, micropreemie

## Abstract

The fetal-to-neonatal transition poses an extraordinary challenge for extremely low birth weight (ELBW) infants, and postnatal stabilization in the delivery room (DR) remains challenging. The initiation of air respiration and the establishment of a functional residual capacity are essential and often require ventilatory support and oxygen supplementation. In recent years, there has been a tendency towards the soft-landing strategy and, subsequently, non-invasive positive pressure ventilation has been generally recommended by international guidelines as the first option for stabilizing ELBW in the delivery room. On the other hand, supplementation with oxygen is another cornerstone of the postnatal stabilization of ELBW infants. To date, the conundrum concerning the optimal initial inspired fraction of oxygen, target saturations in the first golden minutes, and oxygen titration to achieve desired stability saturation and heart rate values has not yet been solved. Moreover, the retardation of cord clamping together with the initiation of ventilation with the patent cord (physiologic-based cord clamping) have added additional complexity to this puzzle. In the present review, we critically address these relevant topics related to fetal-to-neonatal transitional respiratory physiology, ventilatory stabilization, and oxygenation of ELBW infants in the delivery room based on current evidence and the most recent guidelines for newborn stabilization.

## 1. Introduction

A successful fetal-to-neonatal transition relies on a series of exquisitely orchestrated physiological events that enable the switch from the placental intervillous gas exchange to lung air respiration [1]. The initiation of breathing movements immediately after birth in term infants is characterized by deep inspiratory diaphragmatic, intercostal, and subcostal muscular contractions that cause a tridimensional expansion of the thoracic cage. Thorax expansion contributes to the creation of a negative transthoracic pressure equal to or greater than −40 cmH_2_O. This is the driving force that will facilitate the extrusion of the fluid filling the conductive airways into the alveolar interstitial space. Lung aeration permits alveolar gas exchange, which subsequently causes a sudden increment of arterial partial pressure of oxygen (PaO_2_) [2,3,4].

Oxygen is a potent vasodilator that contributes to pulmonary arterial vessel dilatation, thus reducing pulmonary vascular resistance [5]. Consequently, there is a significant increase in pulmonary blood flow from 138 to 245 mL/kg/min [1]. In addition, increased PaO_2_ favors the closure of the ductus arteriosus [1]. Moreover, enhanced preload of both the right and the left ventricles will mechanically contribute to the closure of the foramen ovale. Thus, in several minutes, the newborn infant switches from a serial to a parallel type of circulation that will persist throughout their entire life [1].

In term newborns, birth asphyxia is the most common condition needing resuscitation [6]. The main intervention under these circumstances will be to provide positive pressure ventilation to restore an effective gas exchange and to supply oxygen and glucose to the central nervous system and myocardium, both of which are essential for the infant’s survival [7]. However, preterm infants who represent approximately 10% of all deliveries worldwide are especially predisposed to difficulties in establishing a regular and effective pattern of respiration immediately after birth. The immaturity of the respiratory drive and lung cytoarchitecture, the lack of surfactant production, the excessive elasticity of the thoracic cage, and the debility of the respiratory musculature often hamper the initiation of the first inspiratory movements, the establishment of an effective clearance of lung fluid, and the establishment of a functional residual capacity (FRC). All these circumstances frequently lead preterm infants, especially very preterm infants, to respiratory insufficiency. Consequently, a substantial proportion of preterm infants will need respiratory support in the first minutes after birth including oxygen supplementation to satisfactorily undergo postnatal stabilization [8].

Only a decade ago, preterm babies needing respiratory support in the first minutes after birth were directly intubated and ventilated [9]. However, the concept of “the first golden minutes” was put forward, aiming to achieve postnatal stabilization by employing the least aggressive approach in babies who were mostly spontaneously breathing and crying and just needed some support to overcome the difficulties inherent to their immaturity. This concept was primarily targeted toward ventilation upon stabilization in the delivery room (DR) [10].

In recent years, delaying cord clamping has become a standard of care in term and preterm infants [11,12]. Evolving pulse oximeter saturation and heart rate (HR) in the first 10 min after birth in healthy term babies with delayed cord clamping significantly differs from Dawson’s nomogram, which constitutes the reference range for the management of oxygen in the DR [13,14]. Initiating ventilation with a patent cord has implied physiological advantages in the experimental setting [15]. This new approach to positive pressure ventilation in the DR is being explored by different research groups [16,17,18,19,20]; however, results are not yet available, therefore caution in the application of this new modality of ventilation should guide our steps and be restricted to randomized controlled trials (RCTs).

Positive pressure ventilation (PPV) in the DR is of paramount importance for the resuscitation of term and the stabilization of preterm infants immediately after birth. The present review article provides a critical update to the recently published experience of ventilation in the DR of extreme preterm defined as newborn infants born at <28 weeks gestation.

## 2. Physiology of the Respiratory Fetal-to-Neonatal Transition

During fetal life, the gas exchange between mother and fetus takes place in the intervillous spaces of the placenta. Increased pulmonary vascular resistance renders pulmonary blood flow almost inexistent. Hence, oxygenated blood is redirected through fetal shunts (ductus venosus, foramen ovale, and ductus arteriosus) to the upper part of the body and especially to the cerebral and coronary circulations, thus avoiding non-perfused lungs that are not responsible for the fetal gas exchange [21]. Hence, oxygenated blood from the placenta is only partially merged with deoxygenated blood from the fetal superior vena cava.

At birth, physiological factors such as the interruption of cord blood flow and the initiation of inspiratory efforts will cause significant hemodynamic changes. A sudden increase in the systemic vascular resistance caused by the interruption of the umbilical arteries’ blood flow contributes to the inversion of flow in the ductus arteriosus and foramen ovale causing their functional and anatomical closure. Simultaneously, the first inspiratory efforts extrude liquid filling the lungs, favor surfactant distribution, prompt lung aeration, and establish an FRC, which triggers a decrease in pulmonary vascular resistance, increases the pulmonary blood flow and oxygenation, and reduces the work of breathing. The establishment of an FRC is the cornerstone of the postnatal adaptation of the newly born infant.

Subsequently, the left heart preload with oxygenated blood increases and closes the septal layer of the foramen ovale. Hence, the sudden postnatal increase in pulmonary blood flow directly depends on lung aeration and oxygenation and is essential to prevent the drop in cardiac output that will occur if cord clamping precedes lung aeration [21].

The fetal-to-neonatal transition establishes parallel-type circulation in which deoxygenated blood reaches the right heart and is directed to low-resistance pulmonary circulation where oxygenation takes place. By contrast, the full return of oxygenated blood from the lungs to the left heart is directed to systemic circulation to provide tissue oxygenation [21].

There are a number of reasons why the physiological fetal-to-neonatal transition can be disrupted in preterm infants [21,22,23]: Surfactant deficit, weaker respiratory drive, periodic respiration or even apneic episodes, low inherent cardiac contractility, limited ability to adjust cardiac output, poorer tolerance to high systemic vascular resistance, and persistence of fetal shunts. In this context, respiratory support in the delivery room may be the key to a successful or failed transition and, finally, to better or worse clinical outcomes.

Figure 1 summarizes the alveolar changes in the fetal-to-neonatal transition. In preterm infants, although crying and spontaneous breathing is often present, the respiratory effort can be weaker than necessary to establish an FRC. Under these circumstances, the application of positive airway pressure will contribute to the generation of a sufficient pressure gradient to extrude the fluid in the lung to the interstitial space. In addition, preterm infants have delayed clearance of lung fluid because of decreased sodium resorption and weaker respiratory drive [24,25]. Either a large pressure gradient (inflating pressure) or a greater duration of inflation may be needed [26].

Of note, fluid accumulated in the alveolar interstitium generates positive pressure against the alveoli, which translates into an increased tendency of alveolar collapse and re-introduction of fluid within the alveoli during the expiratory phase. These circumstances are aggravated by a lack of surfactants in premature infants. Therefore, a positive end expiratory pressure (PEEP) or a continuous positive airway pressure (CPAP) has to be applied as respiratory support to premature newborn infants to prevent lung edema and/or atelectasis and alveolar collapse [26].

During fetal life, the larynx offers resistance to the efflux of lung liquid and, in contrast to postnatal life, when apnea occurs in the fetus, the glottis remains closed in an attempt to prevent the loss of intrapulmonary fluid that is essential for fetal lung growth [27]. Recent imaging studies in a rabbit model have shown that the onset of spontaneous breathing may be the key step for this reflex to change and, therefore, for the glottis to remain open. Hence, when ventilation fails in an apneic newborn infant, an airway obstruction secondary to a persistence of the fetal glottis reflex may be the cause. In this situation, the focus should be shifted toward stimulating breathing and avoiding the cause of apnea by standard maneuvers such as cutaneous stimulation or rubbing the back [27].

## 3. Oxygen Supplementation in the Delivery Room

### 3.1. Oxygen in the Fetal-to-Neonatal Transition

The initiation of breathing immediately after birth boosts the availability of oxygen in the newly born infant. The PaO_2_ rises from 30–40 mmHg (4–5.3 kPa) in utero to 70–80 mmHg (9.3–10.6 kPa) ex utero in a few minutes. Oxygen saturation, which fluctuates from 45–55% in the fetus, rapidly increases to 85–95% in the first 10 min after birth [28]. It is vital for the survival of very preterm infants (<32 weeks of gestation) to rapidly achieve optimal oxygenation in the first minutes after birth when the energy demands exponentially increase and gas exchange through the placenta is interrupted. As shown in Figure 2, a careful balance in postnatal oxygenation is necessary because oxygen in excess causes oxidative stress and harms cellular structures. On the other hand, postnatal hypoxemia is associated with increased mortality and/or intra-peri-ventricular hemorrhage (IPVH) [29].

Current guidelines recommend preductal oxygen saturation (SpO_2_) monitoring with a pulse oximeter and keeping SpO_2_ within recommended ranges after birth titrating FiO_2_ accordingly [7,30]. SpO_2_ targets for the first 10 min after birth were based on the percentiles reference range put forward by Dawson JA et al. [13]. Dawson’s reference range was built by merging three databases of newborn infants who did not need resuscitation in the delivery room (DR). Only 30% of these babies were preterm and most of them were late preterm [13]. Resuscitation guidelines (4) generally use the 25th percentile of Dawson’s nomogram as a reference for SpO_2_ as the lower acceptable value. SpO_2_ considered “safe” evolves from levels >60–65% in 2 min to >80% in 5 min [7,31]. The recommendation of delaying cord clamping in recent guidelines [7] changed the reference range values for HR and SpO_2_ in the first 10 min after birth. Ashish KC et al. [32] randomized 1510 women with fetal HR ≥ 100 ≤ 160 bpm and gestational age (GA) ≥ 33 weeks to cord clamping ≤60 s and ≥180 s after birth. SpO_2_ was 18% higher by 1 min, 13% higher by <5 min, and 10% higher by 10 min in babies in the group of prolonged cord clamping. Padilla et al. [14] built SpO_2_ and HR curves in healthy-term newborns with delayed cord clamping for a median of 111 s. Compared with Dawson’s curves, they found significantly higher SpO_2_ and HR already in the first minute after birth. Thus, the median and IQR of SpO_2_ (%) at 1, 5, and 10 min after birth were 77 (68–85), 94 (90–96), and 96 (93–98), respectively. HR (beats per minute) median and IQR at 1, 5, and 10 min after birth were 148 (84–170), 155 (143–167), and 151 (142–161). Badurdeen et al. [16] stabilized preterm infants using physiologically based cord clamping (PBCC). PBCC is defined as clamping the cord after establishing lung aeration. Preterm infants at ≥32 + 0 weeks GA were randomized to either early cord clamping (ECC) or PBCC either using PPV or effective spontaneous breathing prior to cord clamping. The median GA for both groups was approximately 39 weeks GA. Cord clamping was performed at a median of 136 s in the PBCC group and 37 s in the ECC group. No differences in HR between 1 and 2 min after birth were evidenced. The percentile chart for SpO_2_ was elaborated with a total of 295 infants >35 weeks GA with delayed cord clamping ≥ 2 min. Out of these, 54.6% were born vaginally and 45.4% by C-section. The median SpO_2_ reached 85% at 4 min and 90% at 5 min.

### 3.2. What Initial FiO_2_ Is Best for Very Preterm Infants in the First Minutes after Birth?

In a systematic review and meta-analysis, Saugstad et cols. [33] showed that the use of room air as compared with 100% oxygen for the resuscitation of asphyxiated term infants significantly reduced mortality. The 2010 international resuscitation guidelines were modified and room air was recommended as the initial FiO_2_ for asphyxiated term infants needing resuscitation [34,35,36].

Concomitantly, a series of clinical studies summarized in Table 1 compared the use of higher vs. lower initial FiO_2_ levels during the stabilization of preterm infants in the delivery room. The results of these studies concluded that it was feasible to stabilize very preterm infants with a lower (<0.3) initial FiO_2_. In addition, the use of lower initial FiO_2_ reduced oxidative stress and inflammation as reflected by specific biomarkers [37,38]. However, in preterm infants in the lower FiO_2_ group, the achievement of target saturation was often delayed and therefore these babies needed an increase in the inspired fraction of oxygen. In 2015, the International Liaison Committee on Resuscitation (ILCOR) guidelines recommended initiating the resuscitation of newborns <35 weeks GA with room air and very preterm infants <32 weeks of gestation with FiO_2_ 0.3, thus discouraging the use of a higher concentration (FiO_2_ 0.65–1.0) [39]. In a recent survey, most NICUs in high-income countries initiated respiratory support in moderate-late preterm with FiO_2_ 0.21 (43%) or 0.3 (36%) [40]. However, only 45% titrated FiO_2_ to targeted SpO_2_. Interestingly, of the 695 respondents, while 90% had pulseoximeters in the DR, only 69% had access to oxygen blenders, rendering oxygen titration impossible [40].

Oei et al. [51], in a randomized controlled non-blinded trial, the TORPIDO trial, compared mortality and clinical outcomes of preterm infants <32 weeks GA initially stabilized with room air or 100% oxygen. Unexpectedly, in a post-hoc analysis, the room-air group showed a significantly increased relative risk of death (RR 3.9; CI 1.1–13.4) in the subgroup of newborns <28 weeks GA. The investigators acknowledged that the trial had been interrupted before achieving the number of patients calculated to achieve the statistical power. Moreover, the study had not been powered for this specific secondary outcome. Notwithstanding, they cautioned against the use of room air in newborns <28 weeks [51].

Lui K et al. [52], employing the Cochrane methodology, aimed to determine whether using a lower (FiO_2_ < 0.4) or higher (FiO_2_ ≥ 0.4) initial oxygen concentration titrated to targeted SpO_2_ improved short- and long-term mortality and/or morbidity. The study included randomized controlled trials but also cluster- and quasi-randomized trials. A total of 10 trials with 914 infants were included. The results did not show any differences in mortality compared to discharge or secondary outcomes such as bronchopulmonary dysplasia (BPD), retinopathy of prematurity (ROP), IPVH, periventricular leukomalacia (PVL), necrotizing enterocolitis (NEC), or persistent ductus arteriosus (PDA). Moreover, no differences in neurodevelopmental disability were assessed at 2 years. However, the quality of the evidence was defined as low due to the high risk of bias and imprecision. The authors concluded that there is uncertainty as to whether using higher or lower initial FiO_2_ in preterm <32 weeks GA targeted to SpO_2_ in the first 10 min after birth has a significant effect on mortality or major morbidities or long-term neurodevelopmental disability at 2 years of age [52].

In 2019, Welsford et al. [53] carried out a systematic review and meta-analysis comparing the clinical outcomes of 5697 preterm infants <35 weeks GA stabilized with an initial lower oxygen concentration (≤50%) vs. a higher oxygen concentration (>50%). No differences were found in short-term mortality, neurodevelopment at two years, or other morbidities such as IVH, BPD, ROP, or NEC among others [53]. It was concluded that there is no benefit or risk in initiating resuscitation with lower or higher FiO_2_ [53].

### 3.3. Long-Term Outcomes and the Initial FiO_2_

Only a few follow-up studies dealing with the long-term influence of the initial FiO_2_ provided during postnatal stabilization in preterm infants have been published (see Table 2). Soraisham et al. [54], in a retrospective cohort study, assessed death and/or neurodevelopmental impairment (NDI) in 1509 preterm infants <29 weeks GA stabilized with FiO_2_ of 0.21, 1.0, or intermediate values in the delivery room. The composite score of death or NDI was not different for either of the three groups. However, in survivors, the adjusted odds for severe NDI were significantly higher in the group resuscitated with 100% oxygen compared with 0.21 (adjusted OR 1.57, 95% CI 1.05, 2.35). It was concluded that the use of 100% oxygen could be a triggering factor for brain inflammation and damage that impaired neurodevelopment in survivors in infancy [54]. In 2011, the Neonatal Resuscitation Program (USA) recommended the switch from an initial FiO_2_ of 1.0 to 0.21 for the stabilization of preterm infants. Kapadia et al. [55], in a retrospective observational study, analyzed the consequences regarding mortality, relevant neonatal morbidities, and NDI in preterm infants of this significant change. The results showed that mortality in the newborn period was not different between groups; however, survivors of the room-air group scored better in the motor composite score of the Bayley III Scale. Moreover, babies in the low-oxygen period spent fewer days on oxygen and had a lower incidence of BPD [55]. Boronat N et al. (23) compared mortality and neurodevelopmental outcomes of extremely preterm infants at the 24-month corrected age assigned to an initial FiO_2_ 0.3 vs. 0.6–0.65 in two randomized controlled and blinded trials. No differences in mortality were established. Moreover, Bayley III scales motor, cognitive and language composites, neurosensorial handicaps, cerebral palsy, or language skills did not show any differences between groups.

### 3.4. SpO_2_ Targets and Oxygen Titration

The initial FiO_2_ should be titrated to achieve targeted SpO_2_s at specific time points using an air–oxygen blender (Figure 3). The most widely employed references in the literature are the recommendations of the 2010 American Heart Association resuscitation algorithm aiming at SpO_2_ 70–75% at 3 min and 80–85% at 5 min [34] and of the European Resuscitation Council’s (ERC) newborn life support algorithm that recommends reaching 70% at 3 min and 85% at 5 min [36]. These targeted values are close to the median values of newborn infants not needing resuscitation. However, evidence-based information regarding the pulse oximetry response to increasing or decreasing FiO_2_ in preterm infants is lacking. A group of experts recommended based on their experience that if SpO_2_ was below the 10th percentile, then FiO_2_ should be increased in 10% increments every 30 s aiming to reach the 25–50th percentile avoiding SpO_2_ higher than 90% because this may be associated with PaO_2_ clearly reaching a toxic value [58].

The target SpO_2_ range recommended for term and preterm infants are similar, and most guidelines [7,59] recommend keeping SpO_2_ above percentile 25 of the reference charts during the first 10 min after birth. For premature newborns, this recommendation poses uncertainty since the percentile SpO_2_ curves were obtained predominantly from healthy-term infants. Preterm infants included were primarily late preterm, and there was a very low representation of preterm infants < 32 weeks [13].

In 2006, Canada changed the resuscitation policy from 100% oxygen for all born babies to 21%. In 2015, Rabi et al. [60], in a retrospective cohort study, compared a historical cohort of preterm babies ≤27 weeks GA before the change in the resuscitation policy (2004–2006) with a cohort after the policy change (2007–2009). They found increased mortality and a higher risk of severe neurological injury in newborns resuscitated initially with room air. During the study period of 2004–2009, the use of pulse oximetry in the DR was not yet standardized, nor were there established target SpO_2_ ranges. Thereafter, reference ranges for SpO_2_ and titration policies were implemented. In fact, a meta-analysis in newborns ≤ 28 weeks [61] found no differences in mortality and morbidity when comparing resuscitation with an initial higher vs. lower oxygen concentration. Intriguingly, they found differences in mortality depending on whether the studies were blinded (favoring low FiO_2_) or unblinded (favoring high FiO_2_). Although the cause of this effect is not known, the authors suggested that the adjustment of FiO_2_ in response to changes in SpO_2_ could be the key to this difference. Hence, in a recent meta-analysis performed in newborns <32 weeks, evolving SpO_2_ in the first 10 min after birth was assessed [62]. Of note, only 25% of newborns reached the target SpO_2_ (≥80–85%) at 5 min. Moreover, not reaching 80% SpO_2_ at 5 min was associated with a higher rate of severe IVH, and the longer the time needed to reach SpO_2_ >80%, the higher risk of death. Newborns with lower GA and lower initial FiO_2_ were more likely to fail to achieve a SpO_2_ of 80% [62]. In an individual patient meta-analysis from three randomized trials comparing higher (>0.6) vs. lower (<0.3) initial FiO_2_ in preterm infants <32 weeks GA, Oei JL et al. [63] found that initial FiO_2_ was not associated with differences in death and/or disability or cognitive scores <85 at 2 years of age. However, SpO_2_ >80% at five minutes was associated with decreased disability/death and cognitive scores >85. It may be concluded that, more so than the initial FiO_2_, it is achieving oxygen saturation above 80% five minutes after birth that reduces mortality and brain damage in very preterm infants.

Oxygen is a potent stimulator of respiratory drive. In experimental studies, it has been shown that hypoxemia induces respiratory depression [64]. Dekker et al. [65] performed a small RCT comparing two groups of newborns < 30 weeks who were resuscitated with 30% (*n* = 24) vs. 100% (*n* = 20) and observed their effect on spontaneous ventilation. They retrieved detailed information on the management of oxygen titration during stabilization and measured oxidative stress biomarkers. They found that the group resuscitated with 100% O_2_ had a better respiratory effort with significantly higher tidal volumes (V_T_), better oxygenation with a shorter duration of hypoxemia, and a shorter duration of ventilation compared to the 30% group, with no increased risk of hyperoxia or oxidative stress [65]. However, the rates of intubation in the DR or at <24 h, IVH grades III, death, or BPD were not different [63]. These results open the door to adequately powered RCTs comparing the use of higher vs. lower initial oxygen concentrations with rapid titration to avoid an oxygen overload. Until then, the results of this study should be treated with caution.

### 3.5. Current Recommendations

Currently, the ILCOR 2020 guidelines [7] recommend the use of lower initial FiO_2_ (0.21–0.30) for newborns < 35 weeks who receive respiratory support at birth (weak recommendation, with very low certainty of the evidence). However, there are some differences between the different guidelines. While the American Heart Association (AHA) [30] recommends an initial FiO_2_ of up to 0.3, the ERC [59]) recommends using a low oxygen concentration depending on GA (0.21 in newborns ≥ 32 weeks, 0.21–0.3 in newborns 28–31 weeks, and 0.3 in newborns < 28 weeks).

## 4. Respiratory Support

Despite the immaturity of the lung, thoracic cage and muscles, and respiratory drive, approximately 80% of very preterm infants (<32 weeks GA) and even extremely preterm <26 weeks GA initiate spontaneous breathing or crying at birth. [66,67]. The ILCOR 2020 guidelines recommend nasally administered continuous positive airway pressure (nCPAP) to provide ventilatory support and establish/maintain lung FRC [7]. The use of tracheal intubation has declined over the last decade in very preterm infants in the first golden minutes and may confer advantages toward survival without major morbidity [68].

In recent years, proactive resuscitation has substantially increased the rate of survival of micro-preemies in the limit of viability (22–24 weeks GA). The ventilatory support strategies employed to enhance morbidity-free survival rates vary considerably among institutions reflecting the lack of evidence-based studies, which hampers the establishment of consensus guidelines as has been acknowledged by different international resuscitation guidelines [7,30,59,69,70].

We approach different aspects of stabilization and resuscitation during the golden minutes and focus especially on the subgroup of premature infants born <25 weeks gestation.

### 4.1. Ventilation in Preterm Infants in the Delivery Room

Immediately after birth, caregivers aim to stabilize preterm infants, independently of their GA, on non-invasive respiratory support [10]. CPAP remains the most widely employed mode of noninvasive respiratory support to establish an FRC and achieve lung recruitment. However, extremely preterm infants frequently require a face mask for intermittent positive pressure ventilation support (IPPV) with PEEP during the initial stabilization phase in the transitional period. The effectiveness of face mask PPV requires choosing the right size to cover the nose and mouth, applying it without excessive pressure to avoid compression of the trigeminal-cardiac reflex, detecting face mask leakage or airway obstruction with a respiratory function monitor (RFM), and repositioning the mask [8,71] (Table 3).

### 4.2. Modalities of Non-Invasive Ventilation: CPAP and PPV Plus PEEP

A general summary of the respiratory approach is depicted in Figure 4. In spontaneously breathing preterm infants CPAP with 6 cmH_2_O and non-breathing newborns, intermittent PPV with an inflation pressure of 20 to 25 cmH_2_O and PEEP of 5 cmH_2_O at a rate of 40 to 60 breaths/min should be provided early after birth [72]. Ideally, pressures should be adjusted according to lung compliance [73] and the patient’s evolving HR and SpO_2_ [9,74,75].

For infants requiring higher supplemental oxygen, the CPAP level may be titrated higher up to 7–8 cm [76]. It has been shown that stepwise increments of PEEP after birth improved the rates of survival and reduced morbidity in preterm infants [77]. Petrillo et al. showed that sustained inflation (SI) followed by nCPAP in the range of 6 to 8 cmH_2_O, administered with the RAM nasal cannula (RAM Nasal Cannula, Neotech Products, Valencia, CA, USA) vs. face mask CPAP of 5 cmH_2_O, resulted in a significant reduction of the intubation rate in the DR [78]. Increasing the pressure in the initial minutes after birth should be carefully considered. Hence, 75% of neonates born at <29 weeks’ GA resuscitated with a T-piece resuscitator (TPR), set with a peak inspiratory pressure (PIP) of 24 and PEEP of 6 cmH_2_O, received V_T_ > 6 mL/kg, which can injure the lungs and contribute to IVH [79].

No optimal CPAP pressure has been established. However, preclinical studies in preterm lambs [80,81] and rabbits [82] compared titrated pressures in a range of 0–12 cmH_2_O and concluded that CPAP levels >10 cmH_2_O improved oxygenation [71], suggesting that uniform lung aeration is best achieved by starting respiratory support with higher PEEP levels. However, higher PEEP levels should be cautiously applied because they could overexpand the lungs and decrease pulmonary blood flow and the breathing rate [71]. In an RCT, the use of CPAP ≥ 8 cmH_2_O during resuscitation significantly increased the rate of pneumothoraces [83], while infants on CPAP levels of 5 to 7 cmH_2_O did not exhibit this [84]. In addition, inadvertent PEEP above the set value should be taken into consideration [73,76].

### 4.3. Type of Devices: T-Piece Resuscitator (TPR), Self-Inflating Bag (SIB), Mechanical Ventilators

It is mandatory that the ventilation devices employed in the DR provide PIP, PEEP, and/or CPAP [8]. The self-inflating bag (SIB) and the T-piece resuscitator (TPR) are the two most common manual ventilation devices employed for respiratory support in DR. However, CPAP is not applicable with SIB. Moreover, the SIB expiratory valves are unreliable for providing PEEP with very low V_T_ [85]. However, the use of SIB is essential for neonatal resuscitation in regions where pressurized gases are not readily available [7].

Experimental studies suggest the benefit of using devices providing controlled levels of PEEP and PIP to assist in the establishment of pulmonary FRC during the transition and reduce lung damage secondary to barotrauma [86,87]. In manikin studies, TPR delivered more consistent PPV and more homogeneous V_T_ than SIB [88,89,90]. In preterm neonates, TPR resulted in better control of PaCO_2_ levels compared to SIB during surfactant administration [74]. In addition, Roehr et al. [91], in a recent systematic review, identified new evidence that pointed towards improved survival, decreased BPD, and fewer intubations at birth in preterm infants stabilized with TPR [92,93].

It is currently unclear which TPR is most effective for applying CPAP at birth. The effect of pressure stability and expiratory resistance was compared with seven CPAP systems in simulated breath profiles. Neopuff (Fisher & Paykel Healthcare, Auckland, New Zealand) and Medijet (Medin CNO Medical Innovations, Puchheim, Germany) had the highest pressure instability and imposed work of breathing. Benveniste (gas-jet valve Dameca, Copenhagen, Denmark, and Prongs Firma H. Mortensen, Randers, Denmark), Hamilton Universal Arabella (Hamilton Medical AG, Bonaduz, Switzerland), and Bubble CPAP (Fisher and Paykel Healthcare, Auckland, New Zealand) showed intermediate results. AirLife (Cardinal Health, Waukegan, IL, USA) and Infant Flow (Viasys Healthcare Respiratory Care, Palm Springs, CA, USA) showed the lowest pressure instability and imposed work of breathing and the lowest decrease in delivered pressure when challenged with a constant leak [94]. A new TPR device (rPAP) that uses a dual-flow ratio valve (fluidic flip) to produce PEEP/CPAP, designed to be used with nasal prongs, showed low imposed work of breathing and kept PEEP at the set value due to inherent TPR device design characteristics, decreasing the rate of intubation or death in the DR [73,95].

Ventilators are commonly used for CPAP delivery and PPV during transport and in the NICU rather than in the DR [9]. However, in the Uppsala University Children’s Hospital (Sweden) and Kitasato University, Kanagawa (Japan), babies in the limit of viability (22–23 weeks GA) are intubated immediately after birth and placed on ventilators with targeted V_T_ avoiding bag and mask ventilation or CPAP [70].

### 4.4. Interfaces for Delivering Mask Non-Invasive Ventilation

Two different types of masks are available for mask PIP or CPAP, anatomic or round masks. O’Donnell et al. found no differences in air leaks between the two different mask types [96]. The effectiveness of the face mask PPV depends on achieving an adequate seal to avoid airway obstruction and mask leak, gently placed to avoid the potential activation of the naso-trigeminal reflex [75,97]. Short binasal prong interfaces typically had greater resistance at the smallest assessed sizes, which could deliver insignificant V_T_ [98]. The experience with its application is limited, but RAM nasal cannula reported a decrease in DR intubation [99]. Small nasal masks can be used as an alternative to binasal prongs generating less intrinsic resistance compared with short binasal prongs [98]. The use of a single nasal tube causes large leaks, more obstruction, delays PPV initiation because of placement, and lower V_T_ and higher requirements for supplemental oxygen compared with the face mask [100]. Effective ventilation decreasing early neonatal mortality and brain injury could be more easily achieved with a supraglottic airway device than with a face mask [101,102]. However, this device has been underutilized due to inappropriate sizes for premature babies, providers’ limited experience, limited knowledge of its functionality, and likely due to a lack of evidence. A high-flow nasal cannula (6–8 L/min) or non-invasive high-frequency oscillation in the DR is not yet recommended until reliable evidence is available [9].

The trigeminal-cardiac reflex and laryngeal closure may reduce the effectiveness of non-invasive respiratory support in premature infants immediately after birth [103]. Experimental studies in rabbits have shown that postnatal hypoxia may lead to the closure of the glottis, rendering PPV ineffective [103]. Moreover, inadequate patency of the glottis reduces the effectiveness of SI [71]. The facial compression caused by the application of a face mask may cause intense bradycardia by inducing a trigeminal-cardiac reflex [104]. Kuypers et al. showed that apnea and/or bradycardia occurred after applying either bi-nasal prongs or a face mask on the face for respiratory support in preterm infants at birth [105]. Cutaneous stimulation and supporting spontaneous breathing could enhance the success of non-invasive ventilation by ensuring that the larynx remains open [106].

### 4.5. Heated and Humidified Gas (HHG)

Heating and humidification of inspired gases for the initial stabilization of preterm infants and during transport to the neonatal unit improve the admission temperature in preterm infants, especially below 28 weeks GA [107,108]. Notwithstanding, recommendations for conditioning gases during newborn stabilization cannot yet be given based on the limited evidence currently available, as has been underscored in the last ILCOR CoSTR survey [7].

### 4.6. Endotracheal Intubation

ILCOR 2020 guidelines [7] recommended the use of CPAP rather than intubation for spontaneously breathing preterm infants with respiratory distress requiring respiratory support in the delivery room. Both RCTs and meta-analyses with high-quality evidence have shown a reduction in the combined outcome of death and BPD when starting treatment with CPAP compared with intubation and ventilation in very preterm infants with respiratory distress [83,84,109,110,111,112,113]. The meta-analysis reported no differences in the individual outcomes of mortality, BPD, pneumothorax, IVH, NEC, or ROP [112].

ERC 2021 guidelines suggested considering intubation in the case of needing surfactant administration in the DR [59]. However, in a recent Cochrane review, surfactant delivery via a thin catheter to spontaneously breathing preterm infants compared with surfactant administration through an endotracheal tube (ETT) was associated with a decrease in the risk of death, BPD, and severe IVH [114].

### 4.7. Resuscitation of Extreme Preterm in the Limit of Viability

Extremely preterm infants (<28 weeks GA) require early prophylactic non-invasive respiratory ventilation (NIV) initiated in the DR [59,66,69]. However, the most extremely preterm infants, also known as micro-preemies (22–24 weeks GA), require immediate intubation and mechanical ventilation at birth followed by prolonged dependency on non-invasive respiratory support and supplemental oxygen therapy [66]. The SUPPORT trial showed more favorable outcomes in infants who received early nCPAP treatment in comparison with surfactant treatment in the DR [84]. However, infants <24 weeks GA were not included in this study because pre-study trials had shown nCPAP failure in 23-week infants, similar to what was reported in previous studies, such as the COIN trial [83]. Consequently, the number of micro-preemies enrolled in clinical trials has been too small to show clear scientific evidence of the relationship between intubation and an increased risk of BPD.

Available evidence of the effectiveness of NIV applied to this category of patients is limited, and optimal CPAP pressure to obtain a stable FRC is not known. Of note, it is very unusual for micro-preemies to breathe spontaneously. Consequently, tracheal intubation and mechanical ventilation in infants born at 22–23 weeks GA immediately after birth has become a standard of care in centers with a proactive attitude towards the treatment of these patients in the limits of viability [95,115,116].

There are minimal data on micro-preemies in relation to the necessary ventilation pressures. On the one hand, some studies show that a peak inflation pressure of 20 cmH_2_O might be too low to effectively recruit the lungs in extremely premature infants [117,118,119]. In contrast, Bhat et al. performed a prospective study in preterm infants <34 weeks GA to assess combinations of inflation pressures and times and the resulting expiratory V_T_ levels using an RFM. Inflation pressure was the key factor producing significantly higher expiratory V_T_, and a longer inflation time was not necessary to increase expiratory V_T_ [118]. Murthy et al. found that only 27% of infants had expiratory V_T_ greater than 4.4 mL/kg, but these V_T_ were measured only for the first five inflations via ETT when adequate V_T_ rarely occurs [117]. On the other hand, RCT in the DR with a fixed initial PIP and settings according to V_T_ [75] show different pressure levels to achieve adequate lung recruitment depending on GA, with PIP less than 20 cmH_2_O. This strategy is also used regularly in Japanese groups with an active attitude toward micropremies [115]. We suggest a set of maneuvers and strategies for the management of premature infants at the limit of viability (22–23 weeks GA) described in Table 4.

### 4.8. Respiratory Function Monitor (RFM)

The effectiveness of mask ventilation can be improved using an RFM. There are different models and brands of RFM but all of them provide continuous real-time information of inspiratory and expiratory graphs, the respiratory rate, V_T_, SpO_2_, or air leaks. This information allows one to detect problems/pitfalls associated with mask ventilation early [75,97,120]. RCTs comparing the use of an RFM in addition to clinical assessment vs. clinical assessment alone during mask ventilation in the DR of infants born <37 weeks’ gestation showed that using an RFM to guide V_T_ delivery might reduce injury and improve outcomes [75,120,121]. In a meta-analysis of three RCTs enrolling 443 infants, the pooled analysis showed no differences in the rates of death before discharge with an RFM vs. no RFM. However, a significant reduction for any brain injury considered a combination of IVH and PVL (RR 0.65 (0.48 to 0.89), *p* = 0.006) and IVH (RR 0.69 (0.50 to 0.96), *p* = 0.03) was assessed in infants receiving PPV with an RFM vs. no RFM. Moreover, these studies reported that fewer infants in the RFM-visible group had expired V_T_ > 8 mL/kg, compared with the no-RFM group [122]. However, there is insufficient evidence to make a recommendation for or against its use [7,123]. Long-term neurological outcomes to assess the efficacy of RFMs during mask ventilation in preterm infants will help to make a strong recommendation for its use in the delivery room. Perhaps its cost-effectiveness and the training requirements precluded the generalized use of RFM [124]. The monitoring of non-invasive ventilation effectiveness (either by capnography or RFM) is becoming more common outside research settings to detect adverse events to be able to reposition the mask and thus decrease them [122,125,126].

## 5. Conclusions

At present, the optimal initial FiO_2_ and how to titrate oxygen during the stabilization of very preterm infants in the delivery room are yet unknown.

Optimizing ventilation to establish a good lung capacity and cutaneous stimulation to trigger spontaneous breathing both contribute to the establishment of effective respiration in the initial minutes after birth.

Despite the initial FiO_2_, titrating oxygen to achieve SpO_2_ targets of 80–85% five minutes after birth seems appropriate to reduce the damage caused by hypoxia or hyperoxia during resuscitation in the DR.

Reference ranges in newborns with deferred, as compared to immediate, cord clamping show differences in SpO_2_ and HR in the initial minutes after birth.

The initiation of ventilation with an intact cord (physiologic-based cord clamping) seems to be a promising strategy to enhance oxygenation and achieve hemodynamic stabilization in the initial minutes after birth; however, until more evidence is available, caution is advised.

The application of optimal strategies to use NIV modalities immediately after birth is important to establish an FRC to reduce the need for intubation, invasive mechanical ventilation, mortality, and BPD.

Implementing a resuscitation bundle involves determining the appropriate size and sealed mask, head repositioning, the opening of the mouth, increasing the pressure when indicated and regulating it depending on the patient’s response and changes in lung compliance, and debriefing after each intubation.

TPR allows accurate PPV with PEEP. There is no current evidence to suggest one interface is better than another. Evidence was insufficient to recommend the use of heated, humidified gases for assisted ventilation.

Feedback devices such as RFM can help detect adverse events.

To reduce unwarranted variability in the care of most extremely preterm infants between 22 and 23 weeks of GA, we propose respiratory support including immediate oral intubation, applying TPR immediately, and avoiding bag ventilation either by mask or ETT, ECG leads, or an ETT secured lip level. This approach could be considered for use in preterm infants of 24 weeks GA.

## Figures and Tables

**Figure 1 children-10-00351-f001:**
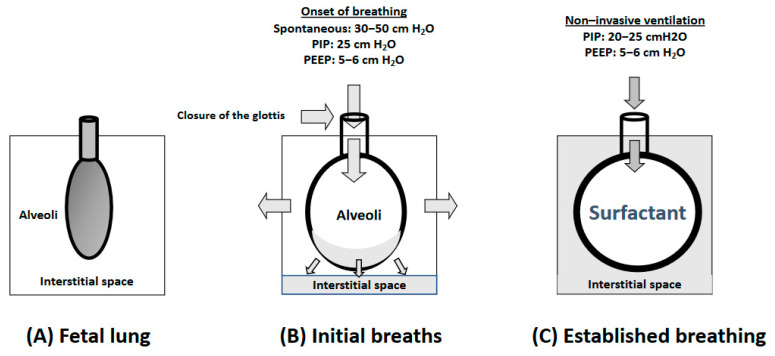
In fetal life, lung and respiratory airways are filled with fluid (**panel** (**A**)). Immediately after birth, intense respiratory efforts generating intense pressures extrude lung fluid to the interstitium, and closure of the glottis and surfactant distribution in the surface of the alveoli contributes to establishing a functional residual capacity (**panel** (**B**)). Ongoing respiratory efforts once the FRC has been established require less positive inspiratory pressures (PIP) and positive-end expiratory pressures (PEEP) to maintain an adequate gas exchange (**panel** (**C**)).

**Figure 2 children-10-00351-f002:**
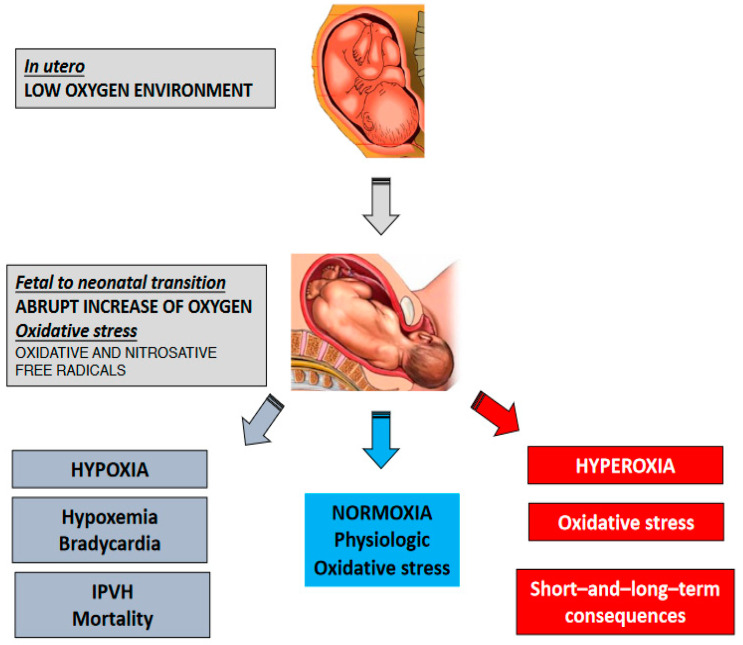
Fetal-to-neonatal transition implies a drastic change to the oxygen provided to the tissue causing physiologic oxidative stress. However, under pathological circumstances, an excess of oxygen can lead to hyperoxia and subsequent pathologic oxidative stress and tissue damage with long and/or short-term consequences. In contrast, low oxygenation can cause hypoxemia, bradycardia, and consequently serious complications such as intra-peri-ventricular hemorrhage (IPVH) and/or death.

**Figure 3 children-10-00351-f003:**
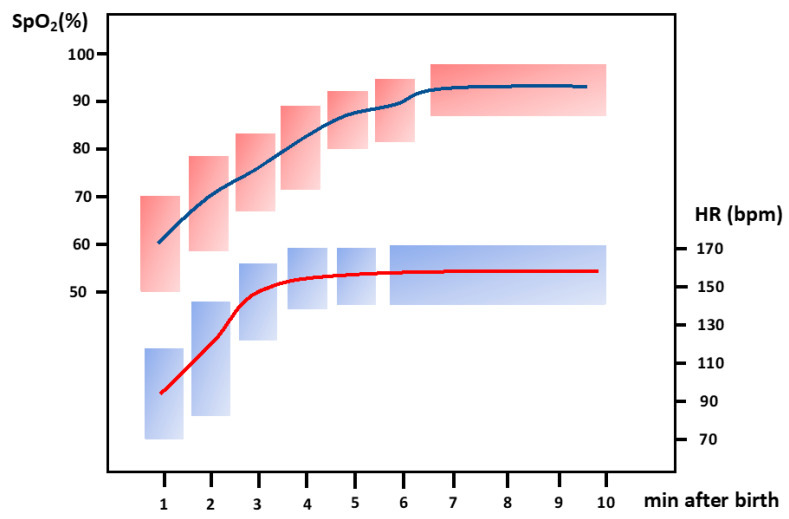
Oxygen saturation (SpO_2_) according to AHA guidelines should be targeted at 70–75% at 3 min and 80–85 at 5 min. Heart rate (HR) should be targeted at >100 bpm in the first 2–3 min after birth. Red and blue rectangles define the normality ranges for SPO_2_ and HR, respectively, at different postnatal timings after birth (Graph modified from reference [28]). Red and blue rectangles define the upper and lower of normality ranges for SPO_2_ and HR, respectively, minute-by-minute in the first 10 min of life.

**Figure 4 children-10-00351-f004:**
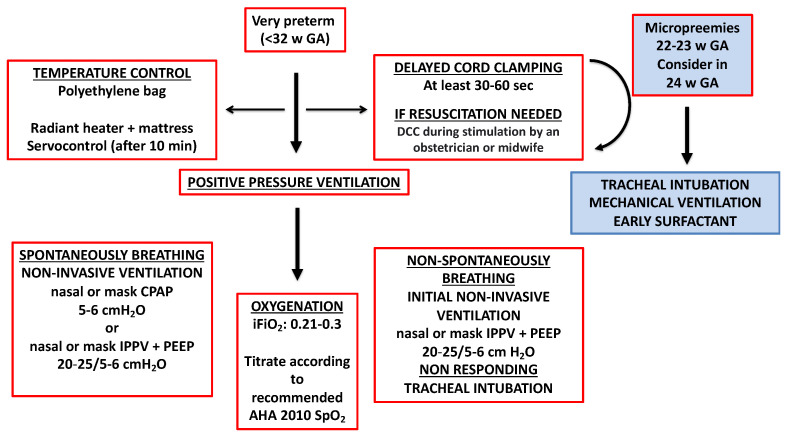
Suggested steps to be followed in very preterm delivery. The main issues in the delivery room are (i) maintaining an adequate body temperature and avoiding hypothermia; (ii) improving hemodynamics stability, delaying cord clamping, or including an obstetrician or midwife in the first seconds of neonatal stimulation; (iii) in spontaneously breathing babies, providing non-invasive ventilation with mask or nasal prongs CPAP, trying to avoid intubation. If the respiratory efforts are insufficient to achieve an adequate FRC, IPPV with PEEP should be provided. (iv) In apneic babies, initial non-invasive ventilation with IPPV and PEEP should be provided. If IPPV and PEEP are not efficient, intubation is required. (v) Initial FiO_2_ of 0.21 to 0.3 should be titrated using an air/oxygen blender according to HR and SPO_2_ response. Modified from Vento et al. Pediatr Respir Rev. (2015) [8]. Abbreviations: GA: Gestational age; w: Week; sec: Second; DCC: Delayed cord clamping; min: Minute; CPAP: Continuous positive airway pressure; IPPV: Intermittent positive pressure ventilation; PEEP: Positive-end expiratory pressure; iFiO_2_: Initial fraction of inspired oxygen; AHA: American Heart Association; SpO_2_: Oxygen saturation.

**Table 1 children-10-00351-t001:** Clinical studies comparing higher or lower initial FiO_2_ for the stabilization of preterm infants in the delivery room.

Reference	Study Design	[Initial FiO_2_]	Objectives	Outcomes
Harling 2005 [41]	RCT GA < 31 weeks No PO; No SpO_2_ targets, *n* = 52	0.5 vs. 1.0	↓ lung inflammation	No significant differences in lung inflammation. It is possible to use lower FiO_2_.
Stola 2005 [42]	Cohort WB < 1500 g PO & target SpO_2_; *n* = 100	Variable vs. 1.0	Viability PaO_2_ at NICU admission	↓ PaO_2_ at NICU admission with lower FiO_2_ Is possible to use less FiO_2_.
Wang 2008 [43]	RCT GA < 32 weeks PO & target SpO_2_; *n* = 41	0.21 vs. 1.0	Viability; SpO_2_ targets: 80–85% at 5 min 85–90% at 7 min	Supplemental O_2_ necessary in 21% group.
Escrig 2008 [44] Vento 2009 [38]	RCT GA < 28 weeks PO & target SpO_2_ *n* = 78	0.3 vs. 0.9	SpO_2_ target 85% at 10 min Oxidative stress; Inflammation	It is possible use less FiO_2_. Less oxidative stress and inflammation in 30% arm.
Dawson 2009 [45]	Cohort GA < 30 weeks PO & target SpO_2_; *n* = 43	0.21 vs. 1.0	Viability SpO_2_ target 90% at 10 min	Supplemental O_2_ is necessary in 21% arm. Is possible to use less FiO_2_
Ezaki 2009 [46]	PO & target SpO_2_ *n* = 44	Variable vs. 1.0	Oxidative stress	↑ oxidative stress in 100% group.
Rabi 2011 [47]	RCT GA < 32 weeks PO & target SpO_2_; *n* = 106	Static 1 vs 0.21 titrate or 1.0 titrate	Viability SpO_2_ target 85–92%	Titrating is more effective than static No differences in timing between the 3 groups to reach the target SpO_2_ range
Armanian 2012 [48]	RCT GA 29–34 weeks PO & target SpO_2_ *n* = 32	0.3 vs. 1.0	SpO_2_ target 85%	It is possible to use less FiO_2_.
Rook 2014 [49]	RCT GA < 32 weeks PO & target SpO_2_ *n* = 193	0.3 vs. 0.65	Major neonatal illness Oxidative stress BPD 36 PMA SpO_2_ target 88–94% at 10 min	30% is as safe as 65%. No differences in oxidate stress or BPD. No differences in oxidative stress biomarkers
Kapadia 2013 [37]	RCT GA 24–34 weeks PO & target SpO_2_ *n* = 193	0.21 vs. 1.0	Oxidative stress Short-term morbidities SpO_2_ target 88–94%	It is possible to use less FiO_2_ Using 21% resulted in less oxidative stress, neonatal morbidities, and need for oxygen supplementation.
Aguar 2013 [50]	RCT GA < 30 weeks PO & target SpO_2_ *n* = 60	0.30 vs. 0.60	Death at 28 days and morbidities SpO_2_ target 88–94% at 10 min	It is possible to use less FiO_2_ No differences in oxidative stress, neonatal morbidities, or mortality.
Oei 2017 [51]	RCT GA < 32 weeks PO & target SpO_2_ *n* = 287	0.21 vs. 1.0	Major disability and death at 2 y SpO_2_ target 65–95% at 5 min and 85–95% until NICU admission	Increased risk of death in infants <28 weeks in the lower FiO_2_ group.

Abbreviations: RCT: Randomized controlled trial; PO: Pulse oximetry; GA: Gestational age; BPD: Bronchopulmonary dysplasia; PMA: Postmenstrual age; SpO_2_: Oxygen saturation; FiO_2_: Inspired fraction of oxygen; WB: Weight birth; NICU: Neonatal intensive care unit; PaO_2_: Arterial partial pressure of oxygen. ↑ is increase and ↓ is decrease.

**Table 2 children-10-00351-t002:** Follow-up of clinical studies of preterm infants stabilized with higher vs. lower initial FiO_2_ in the delivery room.

	Study Design	[Initial FiO_2_]	Objectives	Neurodevelopmental Evaluation Test	Outcomes
Boronat [56]	RCT GA ≤ 32 weeks Pulseoximetry Target SPO_2_ *n* = 206	0.3–0.6	Outcome at 24 months postmenstrual age (PMA)	Bayley III	No differences
Soraisham [54]	Retrospective cohort GA ≤ 28 weeks *n* = 1509	0.21–1.0	Outcome at 18–21 months PMA	Bayley III	Severe NDI in survivors was significantly higher in the 100% oxygen group
Kapadia [55]	Retrospective cohort GA ≤ 28 weeks *n* = 199	0.21–1.0	Outcome at 22–26 months PMA	Bayley III	No differences
Thamrin [57]	RCT GA < 32 weeks *n* = 215	0.21–1.0	Death or NDI at 24 months PMA	Bayley III	No differences SpO_2_ < 80% were more likely to die or to have NDI

Abbreviations: RCT: Randomized controlled trial; SPO_2_: Oxygen saturation; GA: Gestational age; PMA: Postmenstrual age; NDI: Neurodevelopmental impairment.

**Table 3 children-10-00351-t003:** Adverse events that reduce the effectiveness of face mask ventilation.

Error in the size of the facemask. Inadequate positioning causing air leak. Excessive compression that causes upper airway obstruction. Excessive facial compression that causes bradycardia due to the activation of the trigeminal-cardiac reflex. Inadvertent glottis closure impedes air entrance in the lower respiratory airways. Volutrauma caused by excessive V_T_. Atelectotrauma caused by insufficient V_T_ Barotrauma caused by excessive positive pressure in the airways. Ineffective ventilation due to inadequate pressure setting.

Abbreviation: V_T_: Tidal volume.

**Table 4 children-10-00351-t004:** Strategies for the management of preterm infants in the limit of viability (22–23 weeks GA) in the golden hour following the suggestions of neonatal centers with greater experience in the treatment of micro-preemies [13,66,70,75,115,116]. Modified from Norman et al. Semin Fetal Neonatal Med. 2022 [27].

Before birth	Assemble designated staff. Brief with attending team. Prepare delivery room. Use checklists. Information to parents.
Thermal care	Plastic bag without drying. Radiant warmer, room temperature 21–25 °C. Humidified, tempered (36–37 °C) gases for ventilation.
Delayed cord clamping [12]	Any gestational age if resuscitation isn’t needed.
Initial ventilatory support at stabilization (while placing ECG leads on the infant’s chest)	- 22 weeks’ GA: intubation. - 23 weeks’ GA: intubation most likely needed. - 24 weeks’ GA: nCPAP, PPV by TPR or ventilator on mask/nasal prongs, DR intubation if needed
ETT size internal diameter/intubation depth	- 22–23 weeks: 2.5 mm (two attempts) or 2.0 mm/5.5 cm at lip. - 24 weeks: 2.5 mm/5.5–6 cm at lip.
Initial PIP set not to exceed 20 cmH_2_O (TPR flow 10 L/min) [70,75]	- 22 weeks’ GA: 18 cmH_2_O - 23 weeks’ GA: 19 cmH_2_O - 24 weeks’ GA: 20 cmH_2_O
If use RFM	set volume targeted 2.5 mL (5 mL/kg), T_i_ set <1 s and RR 60/min guide by CO_2_et
Initial FiO_2_ [13]	0.3 Titration to achieve SpO_2_ targets of 80–85% at 5 min
Suggested first intended ventilatory settings [70,75,115]	CV: PIP = 20–22 cmH_2_O, PEEP = 5 cmH_2_O, backup frequency = 40–60/min, V_T_ = 4–6 mL/kg. HFOV + VT_hf_ 1 mL: MAP = 10–12 cmH_2_O, frequency = 14–15 Hz, initial amplitude 40–50 cmH_2_O *, I:E 1:2 (1:1 if >15 Hz). Saturation limits: 90–92%.
Surfactant administration	within 2 h of life
After birth	Debriefing with the attending team

CV = conventional ventilation, ETT = endotracheal tube, FiO_2_ = Fraction of inspired oxygen, GA = gestational age, HFOV = High Frequency Oscillatory Ventilation, TPR= T-piece resuscitation, MAP = mean airway pressure, nCPAP = Continuous positive airway pressure administered nasally, PEEP = positive end-expiratory pressure, PIP= peak inspiratory pressure, PPV = Positive pressure ventilation, RR = Respiratory rate, T_i_ = inspiratory time, V_T_ = tidal volume, VT_hf_ = volume guarantee in high frequency oscillator ventilation. I:E = inspiration expiration ratio. * After recruitment, delimit 15–20 cmH_2_O above what is necessary to reach set VT_hf_.

## Data Availability

Not applicable.

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
