# Peer review of "The Respiratory Management of the Extreme Preterm in the Delivery Room"

_children, 2023, doi:10.3390/children10020351_

Round 1
Reviewer 1 Report
In their review authors critically address relevant topics such as optimal initial inspired fraction of oxygen, target saturations in the first golden minutes, oxygen titration to achieve desired stability saturation and heart rate values and physiologic based cord clamping of ELBW infants in the delivery room based on current evidence and in the most recent guidelines for newborn stabilization.
The paper is well written and it is valuable of publication.
Below my only suggestions:
row 80: authors stated “This new approach to positive pressure ventilation in the DR is being explored by different research groups [14–17]”; you can consider to add also another research group who published a pilot study on this topic (Pratesi S, Montano S, Ghirardello S, Mosca F, Boni L, Tofani L, Dani C. Placental Circulation Intact Trial (PCI-T)-Resuscitation With the Placental Circulation Intact vs. Cord Milking for Very Preterm Infants: A Feasibility Study. Front Pediatr. 2018 Nov 27;6:364. doi: 10.3389/fped.2018.00364).
row 105: please add a dot at the end of the sentence.
Figure 2: under “Oxidative stress” appear not so comprehensible symbols next to O2, OH, NO and NOO. Please correct/clarify
Row 176: please correct “data bases” in databases
Row 182: authors stated that “Ashish et al [12,29] randomized 1510…”. Ashish et al corresponds to only citation number 29, please delete 12.
Row 217: authors stated “In a recent survey, most NICUs in HIC initiated..”. It is better to specify what is HIC (high income countries)
Table 1: please correct “Objetive” in Objective
Figure 3: the figure is not so clear. Please add some explanation about curves and red and blu rectangles
Row 414: authors stated “However, higher PEEP should be could cautiously applied…”. Please delete “could”.
Row 467: “..airway device than with a face mask, [103,104].” Please delete the comma before dot.
Row 589: “…interfase is better than another.” Please correct “interfase”
Author Response
In their review authors critically address relevant topics such as optimal initial inspired fraction of oxygen, target saturations in the first golden minutes, oxygen titration to achieve desired stability saturation and heart rate values and physiologic based cord clamping of ELBW infants in the delivery room based on current evidence and in the most recent guidelines for newborn stabilization.
The paper is well written and it is valuable of publication.
Answer:
We thank the reviewer for his positive evaluation of our paper.
Below my only suggestions:
row 80: authors stated “This new approach to positive pressure ventilation in the DR is being explored by different research groups [14–17]”; you can consider to add also another research group who published a pilot study on this topic (Pratesi S, Montano S, Ghirardello S, Mosca F, Boni L, Tofani L, Dani C. Placental Circulation Intact Trial (PCI-T)-Resuscitation With the Placental Circulation Intact vs. Cord Milking for Very Preterm Infants: A Feasibility Study. Front Pediatr. 2018 Nov 27;6:364. doi: 10.3389/fped.2018.00364).
Answer:
According to the reviewer’s suggestion we have included the reference: Pratesi S et al. Front Pediatr 2018.
row 105: please add a dot at the end of the sentence.
Answer
We have corrected this sentence according to the reviewer’s indication.
Figure 2: under “Oxidative stress” appear not so comprehensible symbols next to O2, OH, NO and NOO. Please correct/clarify
Answer
As requested by the reviewer, we have simplified the text of Figure 2. It reads now: Oxidative and Nitrosative Stress Radicals.
Row 176: please correct “data bases” in databases
Answer
We have corrected this sentence according to the reviewer’s indication.
Row 182: authors stated that “Ashish et al [12,29] randomized 1510…”. Ashish et al corresponds to only citation number 29, please delete 12.
Answer
According to the reviewer’s indication we have deleted citation #12.
Row 217: authors stated “In a recent survey, most NICUs in HIC initiated..”. It is better to specify what is HIC (high income countries)
Answer
We have worded the acronym HIC as “high income countries” to facilitate readability.
Table 1: please correct “Objetive” in Objective
Answer
We have corrected text in table 1 as indicated by the reviewer.
Figure 3: the figure is not so clear. Please add some explanation about curves and red and blu rectangles
Answer
Following the reviewer’s suggestion, we have expanded the legend of Figure 3 to clarify the meaning of the red and blue rectangles. The legend of Figure 3 reads now:
“Oxygen saturation (SpO2) according to AHA guidelines should be targeted at 70-75% at 3 min and 80-85 at 5 min. Heart rate (HR) should be targeted > 100 bpm in the first 2 – 3 min after birth. Red and blue rectangles define the normality ranges for SpO2 and HR, respectively, at different postnatal timing.”
Row 414: authors stated “However, higher PEEP should be could cautiously applied…”. Please delete “could”.
Answer
We have modified the text according to the reviewer’s indication.
Row 467: “..airway device than with a face mask, [103,104].” Please delete the comma before dot.
Answer
We have modified the text according to the reviewer’s indication.
Row 589: “…interfase is better than another.” Please correct “interfase”
Answer
We have modified the text according to the reviewer’s indication.
Reviewer 2 Report
This manuscript describes the respiratory management of the extreme preterm in the delivery room. It is not clearly reviewed and structured. I've attached the PDF file with comments and highlights. Thanks.

Author Response
This manuscript describes the respiratory management of the extreme preterm in the delivery room. It is not clearly reviewed and structured. I've attached the PDF file with comments and highlights. Thanks.
Row 75: Delayed cord clamping should be performed when resuscitation is not required.
And there was not reviewed the topic of delayed cord clamping in reference 7.
Answer
We agree with the reviewer. Accordingly, we have introduced 2 references (#11, #12) that refer to delayed cord clamping.
Fig2: IPVH
Answer
IPVH is the acronym for Intra-Peri-Ventricular-Hemorrhage. We have added it in the legend of Figure 2.
Row 182:
Answer
Following the reviewer’s suggestion, we have modified the description of the heart rate. It reads now: “HR ³ 100 £ 160 bpm “
Row 187: HRs in the first minutes after birth. Nos propone: Please, consider making a change; first->1”.
Answer
The sentence in row 187 refers to the difference in SpO2 and HR in term babies with delayed cord clamping as compared to babies with immediate cord clamping. The HR especially is significantly higher already in the first minute after fetal expulsion. We have slightly modified the text which reads now:
“Compared with Dawson's curves, they found significantly higher SpO2 and HR already in the first minute after birth.”
Row 225: Please, describe the more precisely. There are not 28 and 29 weeks GA infants”.
Answer
We agree with the reviewer. The sentence has been omitted. It made reference to the fact that mortality differences were only present in babies <28 weeks of gestation and that no babies >29 weeks died in either group (higher or lower FiO2).
Table 1: Please, describe the more precisely. Insufficient Table 1 information.
Answer
According to the reviewer’s suggestion we have modified the description of Table 1. It reads now:
“Clinical studies comparing higher or lower initial FiO2 for the stabilization of preterm infants in the delivery room.”
Table 1. What is the 'arm' mean? If it is abbreviation, you should describe the full term in advance.
Answer
Following the reviewer’s suggestion, we have change “arm” per “group”. The sentences refers to the group receiving room air.
Row 236: Please, describe more precisely. There is no group of FiO2 0.4.
Answer
According to the reviewer’s indication we have changed the text. It reads now:
“Lui K et al [52], employing the Cochrane methodology, aimed to determine whether using lower (FiO2 <0.4) or higher (FiO2 ³0.4) initial oxygen concentration titrated to targeted SpO2 improved short-and long-term mortality and/or morbidity.”
Row 260: This sentence is different from the content of Table 2.
Answer
The results of the study by Soraisham et al., underscored significant differences between higher and lower FiO2 groups. We have corrected the sentence in Table 2. In the column of “Outcomes” it reads now (Soraisham ref 54): “Severe NDI in survivors was significantly higher in the 100% oxygen group.”
Table 2: Please, describe more precisely the contents of Table 2.
Answer
According to the reviewer’s indication we have modified the title of Table 2. It reads now: “Follow up of clinical studies of preterm infants stabilized with higher versus lower initial FiO2 in the delivery room.”
Figure 3. Please, describe more precisely about the figure. There is no explanation for the blue line, black line, and so on. And add the reference.
Answer
According to the reviewer’s indication we have modified the legend of figure 3. It reads now:
“Oxygen saturation (SpO2) according to AHA guidelines should be targeted at 70-75% at 3 min and 80-85 at 5 min. Heart rate (HR) should be targeted > 100 bpm in the first 2 – 3 min after birth. Red and blue rectangles define the normality ranges for SpO2 and HR, respectively, at different postnatal timings (< 3 minutes, and 3 to 10 minutes after birth). (Reference [28]).”
Row 300: What does p25 mean? It needs to describe.
Answer
We have modified the text and substitute p25 by percentile 25. It reads now: “Most guidelines recommend keeping SpO2 at percentile 25 of the reference charts during the first 10 minutes after birth”.
Row 341: The two studies did not have the same protocol although similar. A further expla-nation is needed to avoid misunderstanding.
Answer
There was a mistake in the last reference that led to confusion. It was not 64 but 66. We are only referring to the study of Dekker et al (66).
Table 3. Does insufficient tidal volume cause volutrauma? Check, please.
Answer
We agree with the reviewer. Accordingly, we have modified the text in Table 3. It reads now:
“Volutrauma caused by excessive VT”
“Atelectotrauma caused by insufficient VT”
Row 392. A description of the manufacturer is required.
Answer
We have added the description of the manufacturer as requested by the reviewer.
It reads now: “RAM Nasal Cannula, Neotech Products, Valencia, CA, USA).”
Figure 4. DCC should be performed when resuscitation is not required. Check, please.
Answer
During stabilization of preterm infants, cord clamping can be deferred in the first seconds while the obstetrician or the midwife performs cutaneous stimulation to induce spontaneous breathing. After 30-60 seconds, if there is no effective respiratory drive, the cord should be clamped, and non-invasive ventilation initiated. Physiologic-based cord clamping cannot be yet recommended as a routine in the DR until results from ongoing studies are available.
Table 4: Please, describe more precisely each content of Table 4 with evidence-based ref-erence. Some contents of the text are different from the contents of Table 4
Answer
According to the reviewer’s suggestion, we have included evidence-based references in table 4.
Row 556. You should explanation the abbreviation.
Answer
VTE refers to exhaled tidal volume. This has been incorporated in the text.
Row 589. There is no description of end-tidal CO2 detectors in the main text.
Answer
Dear reviewer, we planned to expand on this issue; however, the length of the manuscript advised us against doing it. We have changed the text accordingly.
Row 595. You need to add the reference.
Answer
We have suppressed this part and subsequently the reference.
Reviewer 3 Report
A well-written and very excellent review manuscript on respiratory management in the delivery room at ELBWIs (or extreme preterm infants). It seems that it can be published if only the minor part below is modified additionally.
1) What is the criterion for extreme preterm used in the title? Please describe in detail in the main text.
2) Please modify the numbers in H2O, ppaO2, etc. with subscripts.
3) In the RFM section (4.8.), please write the specific type of RFM currently used (including company name, etc.).
4) The following references are duplicated. Please revise the references.
34. Kapadia, V.S.; Chalak, L.F.; Sparks, J.E.; Allen, J.R.; Savani, R.C.; Wyckoff, M.H. Resuscitation of Preterm Neonates With Limited 687 Versus High Oxygen Strategy. Pediatrics 2013, 132, e1488–e1496, doi:10.1542/peds.2013-0978.
48. Kapadia, V.S.; Chalak, L.F.; Sparks, J.E.; Allen, J.R.; Savani, R.C.; Wyckoff, M.H. Resuscitation of Preterm Neonates with Limited 723 versus High Oxygen Strategy. Pediatrics 2013, 132, e1488-1496, doi:10.1542/peds.2013-0978.
10. Vento, M.; Cheung, P.-Y.; Aguar, M. The First Golden Minutes of the Extremely-Low-Gestational-Age Neonate: A Gentle Ap-623 proach. Neonatology 2009, 95, 286–298, doi:10.1159/000178770.
70. Vento, M.; Cheung, P.-Y.; Aguar, M. The First Golden Minutes of the Extremely-Low-Gestational-Age Neonate: A Gentle Ap-785 proach. Neonatology 2009, 95, 286–298, doi:10.1159/000178770.
8. Vento, M.; Lista, G. Managing Preterm Infants in the First Minutes of Life. Paediatr Respir Rev 2015, 16, 151–156, 619 doi:10.1016/j.prrv.2015.02.004.
72. Vento, M.; Lista, G. Managing Preterm Infants in the First Minutes of Life. Paediatric Respiratory Reviews 2015, 16, 151–156, 789 doi:10.1016/j.prrv.2015.02.004.
9. Weydig, H.; Ali, N.; Kakkilaya, V. Noninvasive Ventilation in the Delivery Room for the Preterm Infant. Neoreviews 2019, 20, 621 e489–e499, doi:10.1542/neo.20-9-e489.
76. Weydig, H.; Ali, N.; Kakkilaya, V. Noninvasive Ventilation in the Delivery Room for the Preterm Infant. NeoReviews 2019, 20, 797 e489–e499, doi:10.1542/neo.20-9-e489.
85. Morley, C.J.; Davis, P.G.; Doyle, L.W.; Brion, L.P.; Hascoet, J.-M.; Carlin, J.B. Nasal CPAP or Intubation at Birth for Very Preterm 818 Infants. N Engl J Med 2008, 358, 700–708, doi:10.1056/NEJMoa072788. 819
114. Morley, C.J.; Davis, P.G.; Doyle, L.W.; Brion, L.P.; Hascoet, J.-M.; Carlin, J.B. Nasal CPAP or Intubation at Birth for Very Preterm 890 Infants. N Engl J Med 2008, 358, 700–708, doi:10.1056/NEJMoa072788.
86. SUPPORT Study Group of the Eunice Kennedy Shriver NICHD Neonatal Research Network; Finer, N.N.; Carlo, W.A.; Walsh, 820 M.C.; Rich, W.; Gantz, M.G.; Laptook, A.R.; Yoder, B.A.; Faix, R.G.; Das, A.; et al. Early CPAP versus Surfactant in Extremely 821 Preterm Infants. N Engl J Med 2010, 362, 1970–1979, doi:10.1056/NEJMoa0911783.
115. SUPPORT Study Group of the Eunice Kennedy Shriver NICHD Neonatal Research; Network Early CPAP versus Surfactant in 892 Extremely Preterm Infants. N Engl J Med 2010, 362, 1970–1979, doi:10.1056/NEJMoa0911783.
77. Zeballos Sarrato G, Sánchez Luna M, Pérez Pérez A, et al. New Strategies of Pulmonary Protection of Preterm Infants in the Delivery Room with the Respiratory Function Monitoring. Amer J Perinatol 2019;:1–9. Doi:10.1055/s-0038-1676828.
126. Zeballos Sarrato, G.; Sánchez Luna, M.; Zeballos Sarrato, S.; Pérez Pérez, A.; Pescador Chamorro, I.; Bellón Cano, J.M. New Strategies of Pulmonary Protection of Preterm Infants in the Delivery Room with the Respiratory Function Monitoring. Amer J Perinatol 2019, 36, 1368–1376, doi:10.1055/s-0038-1676828.
5) Please revise the reference 78 according to the the submission guidelines.
78. Weiner GMZJ, Kattwinkel J, Eds. Textbook of Neonatal Resuscitation. 7th Ed. Itasca, IL: American Academy of Pediatrics and American 801 Heart Association; 2016;
Author Response
A well-written and very excellent review manuscript on respiratory management in the delivery room at ELBWIs (or extreme preterm infants). It seems that it can be published if only the minor part below is modified additionally.
1) What is the criterion for extreme preterm used in the title? Please describe in detail in the main text.
Answer
According to the reviewer’s suggestion we have modified the last sentence (lines 85-88) of the Introduction. It reads now:
“The present review article provides a critical update to recently published experience in the ventilation in the DR of extreme preterm, defined as newborn infants born at < 28 weeks’ gestation. “
2) Please modify the numbers in H2O, ppaO2, etc. with subscripts.
Answer
According to the reviewer’s indication we have introduced subscripts along the text.
3) In the RFM section (4.8.), please write the specific type of RFM currently used (including company name, etc.).
Answer
In relation to the respiratory function monitors, we have listed all the monitors that are currently available in the market and employed in the NICU. There is no one single monitor that is universally employed. Thus, we have modified the text (lines 547-549) and it reads now:
“There are different models and brands of RFM but all of them provide continuous real time information of inspiratory and expiratory graphs, respiratory rate, VT, SpO2, or air leak.”
.
4) The following references are duplicated. Please revise the references.
- Kapadia, V.S.; Chalak, L.F.; Sparks, J.E.; Allen, J.R.; Savani, R.C.; Wyckoff, M.H. Resuscitation of Preterm Neonates With Limited 687 Versus High Oxygen Strategy. Pediatrics 2013, 132, e1488–e1496, doi:10.1542/peds.2013-0978.
- Kapadia, V.S.; Chalak, L.F.; Sparks, J.E.; Allen, J.R.; Savani, R.C.; Wyckoff, M.H. Resuscitation of Preterm Neonates with Limited 723 versus High Oxygen Strategy. Pediatrics 2013, 132, e1488-1496, doi:10.1542/peds.2013-0978.
- Vento, M.; Cheung, P.-Y.; Aguar, M. The First Golden Minutes of the Extremely-Low-Gestational-Age Neonate: A Gentle Ap-623 proach. Neonatology 2009, 95, 286–298, doi:10.1159/000178770.
- Vento, M.; Cheung, P.-Y.; Aguar, M. The First Golden Minutes of the Extremely-Low-Gestational-Age Neonate: A Gentle Ap-785 proach. Neonatology 2009, 95, 286–298, doi:10.1159/000178770.
- Vento, M.; Lista, G. Managing Preterm Infants in the First Minutes of Life. Paediatr Respir Rev 2015, 16, 151–156, 619 doi:10.1016/j.prrv.2015.02.004.
- Vento, M.; Lista, G. Managing Preterm Infants in the First Minutes of Life. Paediatric Respiratory Reviews 2015, 16, 151–156, 789 doi:10.1016/j.prrv.2015.02.004.
- Weydig, H.; Ali, N.; Kakkilaya, V. Noninvasive Ventilation in the Delivery Room for the Preterm Infant. Neoreviews 2019, 20, 621 e489–e499, doi:10.1542/neo.20-9-e489.
- Weydig, H.; Ali, N.; Kakkilaya, V. Noninvasive Ventilation in the Delivery Room for the Preterm Infant. NeoReviews 2019, 20, 797 e489–e499, doi:10.1542/neo.20-9-e489.
- Morley, C.J.; Davis, P.G.; Doyle, L.W.; Brion, L.P.; Hascoet, J.-M.; Carlin, J.B. Nasal CPAP or Intubation at Birth for Very Preterm 818 Infants. N Engl J Med 2008, 358, 700–708, doi:10.1056/NEJMoa072788. 819
- Morley, C.J.; Davis, P.G.; Doyle, L.W.; Brion, L.P.; Hascoet, J.-M.; Carlin, J.B. Nasal CPAP or Intubation at Birth for Very Preterm 890 Infants. N Engl J Med 2008, 358, 700–708, doi:10.1056/NEJMoa072788.
- SUPPORT Study Group of the Eunice Kennedy Shriver NICHD Neonatal Research Network; Finer, N.N.; Carlo, W.A.; Walsh, 820 M.C.; Rich, W.; Gantz, M.G.; Laptook, A.R.; Yoder, B.A.; Faix, R.G.; Das, A.; et al. Early CPAP versus Surfactant in Extremely 821 Preterm Infants. N Engl J Med 2010, 362, 1970–1979, doi:10.1056/NEJMoa0911783.
- SUPPORT Study Group of the Eunice Kennedy Shriver NICHD Neonatal Research; Network Early CPAP versus Surfactant in 892 Extremely Preterm Infants. N Engl J Med 2010, 362, 1970–1979, doi:10.1056/NEJMoa0911783.
- Zeballos Sarrato G, Sánchez Luna M, Pérez Pérez A, et al. New Strategies of Pulmonary Protection of Preterm Infants in the Delivery Room with the Respiratory Function Monitoring. Amer J Perinatol 2019;:1–9. Doi:10.1055/s-0038-1676828.
- Zeballos Sarrato, G.; Sánchez Luna, M.; Zeballos Sarrato, S.; Pérez Pérez, A.; Pescador Chamorro, I.; Bellón Cano, J.M. New Strategies of Pulmonary Protection of Preterm Infants in the Delivery Room with the Respiratory Function Monitoring. Amer J Perinatol 2019, 36, 1368–1376, doi:10.1055/s-0038-1676828.
Answer
We thank the reviewer for adverting us of the duplicated references. Accordingly, we have eliminated the duplicated references and re-numbered the remaining.
5) Please revise the reference 78 according to the the submission guidelines.
- Weiner GMZJ, Kattwinkel J, Eds. Textbook of Neonatal Resuscitation. 7th Ed. Itasca, IL: American Academy of Pediatrics and American 801 Heart Association; 2016;
Answer
According to the reviewer’s indication we have modified the reference #78 (now 77) following the submission guidelines.